# Anti-Aging Drugs and the Related Signal Pathways

**DOI:** 10.3390/biomedicines12010127

**Published:** 2024-01-08

**Authors:** Nannan Du, Ruigang Yang, Shengrong Jiang, Zubiao Niu, Wenzhao Zhou, Chenyu Liu, Lihua Gao, Qiang Sun

**Affiliations:** 1Frontier Biotechnology Laboratory, Beijing Institute of Biotechnology, Beijing 100071, China; dunannan14@163.com (N.D.); yrgang@163.com (R.Y.); niuzubiao@126.com (Z.N.); zwzaaggaga@163.com (W.Z.); lcyyyy0000@163.com (C.L.); gaolihua0043@163.com (L.G.); 2Research Unit of Cell Death Mechanism, 2021RU008, Chinese Academy of Medical Science, Beijing 100071, China; 3Nanhu Laboratory, Jiaxing 314002, China; 4The Meta-Center, 29 Xierqi Middle Rd, Beijing 100193, China; jiangshengrong13@163.com; 5Department of Oncology, Beijing Shijitan Hospital, Capital Medical University, Beijing 100038, China

**Keywords:** aging, anti-aging drugs, rapamycin, metformin, acarbose, NAD^+^, lithium, NSAIDS, signal pathway

## Abstract

Aging is a multifactorial biological process involving chronic diseases that manifest from the molecular level to the systemic level. From its inception to 31 May 2022, this study searched the PubMed, Web of Science, EBSCO, and Cochrane library databases to identify relevant research from 15,983 articles. Multiple approaches have been employed to combat aging, such as dietary restriction (DR), exercise, exchanging circulating factors, gene therapy, and anti-aging drugs. Among them, anti-aging drugs are advantageous in their ease of adherence and wide prevalence. Despite a shared functional output of aging alleviation, the current anti-aging drugs target different signal pathways that frequently cross-talk with each other. At present, six important signal pathways were identified as being critical in the aging process, including pathways for the mechanistic target of rapamycin (mTOR), AMP-activated protein kinase (AMPK), nutrient signal pathway, silent information regulator factor 2-related enzyme 1 (SIRT1), regulation of telomere length and glycogen synthase kinase-3 (GSK-3), and energy metabolism. These signal pathways could be targeted by many anti-aging drugs, with the corresponding representatives of rapamycin, metformin, acarbose, nicotinamide adenine dinucleotide (NAD^+^), lithium, and nonsteroidal anti-inflammatory drugs (NSAIDs), respectively. This review summarized these important aging-related signal pathways and their representative targeting drugs in attempts to obtain insights into and promote the development of mechanism-based anti-aging strategies.

## 1. Introduction

With the improvement of life quality and medical support, the lifespan of people around the world keeps increasing. It was estimated that the world’s population over 60 years old may double by 2050 (Figure 1) [1]. The change will increase the incidence of age-related diseases, particularly chronic diseases, in the elderly [2].

de Magalhães et al. pointed out that aging contributed to a variety of diseases leading to the loss of function, such as cardiovascular diseases, cancer, neurodegenerative diseases, and major depressive disorder [3,4,5]. The incidence of these diseases in older adults is also expected to increase substantially in the coming decades [6].

Along with the emerging aging issue, multiple anti-aging strategies were developed. Tang et al. found that dietary restriction (DR) could affect the health of flies and worms and prolong their lifespan [7]. Soultoukis et al. and Piper et al. indicated that DR might help select mutations in nutrient signal pathways that extend lifespan [8,9].

**Figure 1 biomedicines-12-00127-f001:**
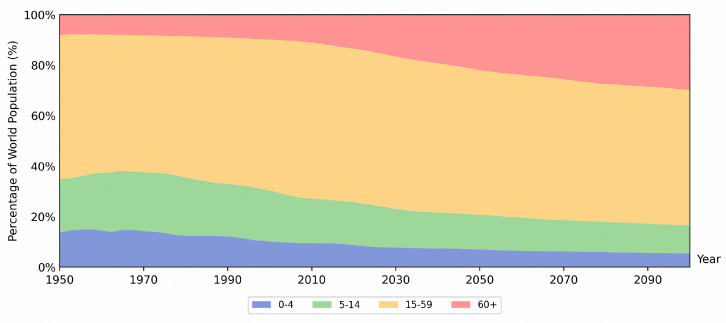
The age distribution predicted by WHO. The population is divided into 4 sections: 0–4 years old, 5–14 years old, 15–59 years old, and 60+ years old. Since 2000, the 0–4 and 5–14 populations start to decrease while the 60+ population is increasing.

Apart from DR, exercise, systemic circulating factors, and gene therapy are also important anti-aging strategies. Hood et al. pointed out that exercise may tune up impaired mitochondrial function to delay or reverse the onset of aging [10]. Using the model of kidney and hippocampus, Huang et al. confirmed that a young blood environment could significantly mitigate renal aging [11]. Castellano et al. showed that human umbilical cord plasma contained plasticity-rich proteins, which target hippocampal dysfunction associated with aging and disease [12]. With regard to gene therapy, DNA double-strand break repair has been shown to be a promising target for anti-aging interventions [13]. 

However, DR and exercise are difficult for the public to stick to, and systemic circulating factors and gene therapy are yet to be approved, which leaves small-molecule drugs a feasible candidate for anti-aging interference [14]. At present, many compounds were proposed to be promising anti-aging candidates that are constantly updated. 

According to the underlying mechanisms, this review summarized six representative types of compounds that target important aging-related signal pathways, including rapamycin, metformin, acarbose, NMN, lithium, and nonsteroidal anti-inflammatory drugs (NSAIDs). These compounds primarily act on the mechanistic target of rapamycin (mTOR) signal pathway, the AMP-activated protein kinase (AMPK) signal pathway, the nutrient signal pathway, the silent information regulator factor 2-related enzyme 1 (SIRT1) signal pathway, the regulation of telomere length and glycogen synthase kinase-3 (GSK-3) related signal pathway, and the energy metabolism-related signal pathway, respectively.

## 2. Rapamycin and the Aging-Related mTOR Signal Pathway (Figure 2)

Rapamycin (Wyeth-Ayerst Laboratories, New York, NY, USA), also called sirolimus, was first isolated in soil samples from Rapa-Nui (Easter Island), and was a natural product of the actinomycete *Streptomyces hygrscopicus* [15]. Rapamycin in complex with the FK binding proteins 12 (FKBP12) functions as a potent inhibitor of a kinase, mechanistic target of rapamycin (mTOR) [15]. Recently, several studies discovered that rapamycin could extend the lifespan in *yeast*, *Caenorhabditis elegans* (*C. elegans*), *Drosophila*, and mice [16,17,18,19,20]. Moreover, it was also confirmed that rapamycin could delay aging in human skin [21].

Researchers found that rapamycin could regulate aging, primarily going through the mTOR signal pathway. They confirmed that S6K’s excessive activation and 4E-BP’s deficiency, as two key downstream effectors of the mTOR signal pathway, would interfere with the effect of rapamycin, which could result a shorter lifespan versus rapamycin used individually in *Drosophila* [18].

Recently, studies found that the mTOR signal pathway was highly related to aging, even to say, the mTOR signal pathway controlled the course of aging [22,23]. Ferrara-Romeo et al. demonstrated that the inhibition of mTOR could increase the survival rate of mice. This phenomenon is mainly due to the deletion of S6 kinase 1, a mTORC1 downstream effector [24]. Additional studies also confirmed that the suppression of mTOR could decrease the senescence-associated secretory phenotype (SASP) in mouse and human cells. Based on this, researchers pointed out that the dysfunction of mTOR would suppress IL1A translation, to decrease NF-κB activity, aiming to modulate the SASP to regulate longevity [25].

In addition, researchers also found that the aging-related regulation of the mTOR signal pathway was diverse and primarily achieved by influencing four important cellular biological processes, including protein homeostasis, mitochondrial respiration, autophagy, and endoplasmic reticulum (ER) stress (Figure 2) [26,27,28,29].

Protein homeostasis is a key way for mTOR signal to regulate longevity. In this process, DAF-16/FoxO and Nrf2/SKN-1, as two kinds of transcription factors, play a key role in aging [30,31,32]. The transcription factor DAF-16/FoxO, by targeting S6K1 and 4E-BP1, inhibits translation and converts cells to a physiological state, thus extending lifespan [30,31,33]. Additionally, researchers also found that when mTOR was inhibited, Nrf2/SKN-1 and DAF-16/FoxO would activate a set of protective genes to regulate longevity, and the special protective genes warrant further investigation.

Mitochondrial respiration is an indispensable way for longevity. It was demonstrated that inhibition of the mechanistic target of rapamycin complex 1 (mTORC1) would decrease PGC-1α’s expression through *YY-1*, which then regulates mitochondrial respiration to extend lifespan. Bonawitzin et al. found that mTOR’s dysfunction could abate the translation of oxidative phosphorylation (OXPHOS) complex, which was encoded by mitochondrial DNA (mtDNA), to influence Sch9p kinase, a key downstream effector of OXPHOS, thus decreasing mitochondrial respiration and increasing longevity [26,27,34,35,36].

Autophagy, discovered by Ohsumi [37], is also an important way of regulating aging [23]. In the aging-related mTOR signal pathway, autophagy is primarily controlled by two molecules, ULK1 and *p73*. Mihaylova et al. pointed out that ULK1, inhibited by mTOR. was a key molecule to promote autophagy in the process of aging [38]. Regarding *p73*, current research revealed that *p73*, the upstream molecule of autophagy, was subjected to negative regulation by mTOR to modulate downstream genes of autophagy, ultimately increasing lifespan [39].

ER stress could regulate longevity in the mTOR signal pathway [29,40]. In mammalian cells, researchers found that hypoxia-inducible factor-1α (HIF1α), one of the mTOR targets, acted in a nutrient-dependent manner to extend lifespan by alleviating the ER stress through inositol-requiring protein-1 (IRE-1) [23,41]. By alleviating ER stress, the unfolded protein response (UPR) would decrease the number of misfolded proteins and inhibit abnormal protein accumulation, thus reducing inflammatory responses to extend lifespan [42].

One point that needs to be emphasized is the above mTOR signal pathways do not function isolatedly. Each signal pathway’s activation or inhibition will affect the others. For example, in the signal pathway of ER stress, HIF-1 not only regulates ER stress, but also inhibits S6K1 [41], which regulates aging by protein homeostasis.

However, the adverse effects of rapamycin constrain its viability as a prospective anti-aging therapy, encompassing interference with the immune system, compromised glucose tolerance, and disrupted balance of lipid metabolism [43,44,45,46,47,48]. Therefore, more clinical investigations are needed to identify rapamycin’s effects in the anti-aging field. In a word, these findings suggested that lifespan extension might be related to multiple signal pathways and multiple gene interactions. The key to extending lifespan is to find the corresponding target, the role of the target in the signal pathway and the inter-pathway relationships.

**Figure 2 biomedicines-12-00127-f002:**
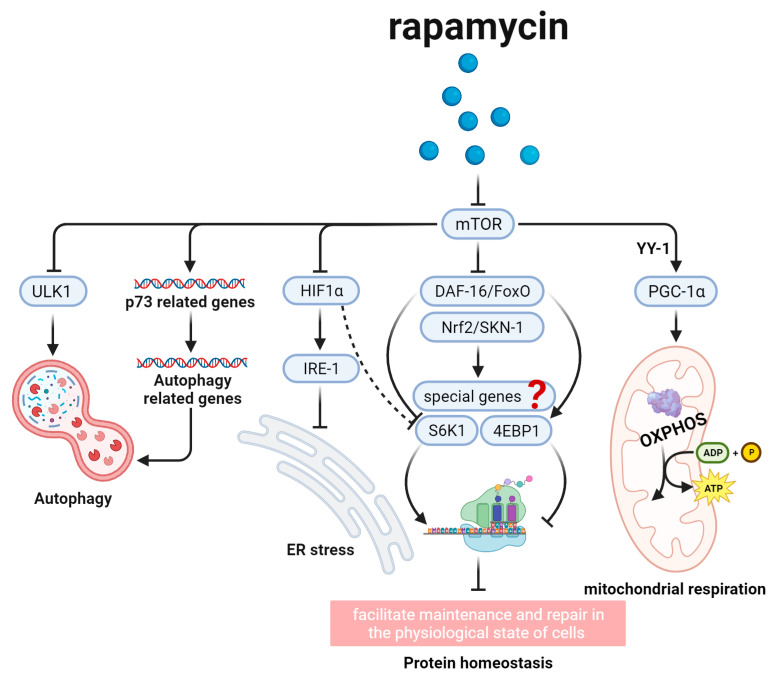
Rapamycin and the aging-related mTOR signal pathway. This figure shows several signal pathways downstream of rapamycin. Rapamycin mainly functions through mTOR and their downstream molecules, such as DAF-16/FoxO and Nrf2/SKN-1, to regulate protein homeostasis, PGC-1α to influence mitochondrial respiration, ULK1 and *p73* to regulate autophagy, and HIF1α to inhibit ER stress, aiming to regulate longevity.

## 3. Metformin and the Aging-Related AMPK Signal Pathway (Figure 3)

Metformin (Merck, Kenilworth, NJ, USA) is a biguanide and is widely used in the treatment of type 2 diabetes mellitus and metabolic syndrome [49,50,51]. Metformin treatment may produce various health benefits, including the reduced risk of cardiovascular disease and cancer [14,52], improved cognitive function in the elderly [53], prolonged survival in patients with diabetes [53,54], and extended lifespan in *yeast*, *C. elegans*, and mice [51,55,56,57].

In 2016, metformin was first developed as an anti-aging drug in a clinical trial, the Targeting Aging with Metformin (TAME) program (https://www.tame-project.org/), because of its potential lifespan-extension effects [58]. Recently, researchers found that metformin’s lifespan-extension effect was mainly through AMP-activated protein kinase (AMPK). In *C. elegans*, the dysfunction of AMPK would eliminate the extension of lifespan induced by metformin. In the study, the researchers found that the deficiency of PEN2, an upstream effector of AMPK, would induce the AMPK’s dysfunction. PEN2 is a target of metformin and intersects the lysosomal glucose-sensing pathway via *ATP6AP1*, thereby activating AMPK [59]. Current studies define four modulations associated with the aging-related AMPK signal pathway, including cellular stress resistance, cell growth, autophagy, and inflammation (Figure 3) [38,60,61,62].

Cellular stress resistance is a crucial point in the aging-related AMPK signal pathway. In cellular stress resistance, researchers showed that AMPK-activation could phosphorylate FOXO [63,64] and inhibit CRTC/CREB, increasing the activity of SirT1, which is mainly by increasing NAD^+^ concentration [65]. Interestingly, SirT1 could directly regulate FOXO and CRTC/CREB to increase cellular stress resistance [66,67], which establish a link among them to contribute toward lifespan-extension. In addition, *TSC1/TSC2*, phosphorylated by AMPK, is also a downstream effector in AMPK signal pathway that negatively regulates mTOR to extend lifespan [68]. Nrf2/SKN-1, as a key effector in the pathway, is regulated by AMPK as well. It was shown that AMPK could stimulate the Nrf2/SKN-1 pathway to regulate longevity [60].

Cell growth is another important feature regulated by the AMPK signal pathway. As described above, mTOR was essential in the protein synthesis of cell growth. When AMPK was activated, mTOR would be inhibited, leading to the inhibition of 4EBP1 and promotion of S6K1 [69], and delayed aging eventually.

Autophagy is also involved in the AMPK signal pathway. ULK1 was essential for the process of autophagosome formation, and was regulated by mTOR [70]. When AMPK is activated, it will promote the phosphorylation of ULK1 [71], contributing to enhanced autophagy. In Parkinson’s disease, a kind of aging-related disease, the mutation of ULK1 phosphorylation site would decrease the events of autophagy, causing an unexpected aging outcome [72].

Inflammation has a vital effect in the aging-related AMPK signal pathway. NF-kB, a classic inflammation molecule, is controlled by AMPK. In mammal cells, AMPK activation promotes the activity of SIRT1 and FOXO to inhibit the NF-kB finally [73]. Thus, the induced inflammation decrease would have an effect in delaying aging.

However, the toxicity of metformin is unignored. Quaile et al. demonstrated that the utilization of metformin at a dosage equal to or greater than 600 mg/kg/day influences the metabolism of glucose and lactate. An even higher dosage equal to or greater than 900 mg/kg/day can result in mortality and the manifestation of clinical symptoms associated with toxicity [74]. Usually, metformin treatment is associated with a prevalence of adverse events, mainly in the gastrointestinal tract, such as diarrhea, nausea, vomiting, flatulence, abdominal pain, and anorexia [75].

**Figure 3 biomedicines-12-00127-f003:**
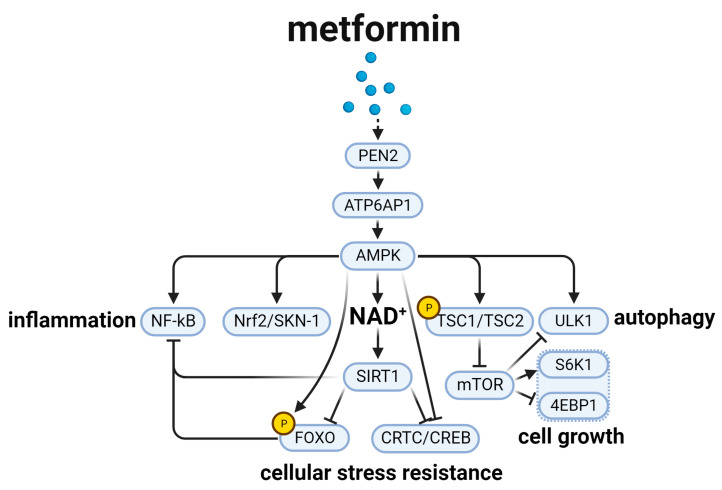
Metformin and the aging-related AMPK signal pathway. Metformin mainly functions through PEN2 to active *ATP6AP1*, thereby inducing AMPK activation. Its downstream molecules include NF-kB to regulate inflammation, Nrf2/SKN-1, SIRT1, and *TSC1/TSC2* to decrease cellular stress resistance, mTOR/S6K1 and mTOR/4EBP to regulate cell growth, ULK1 to activate autophagy.

## 4. Acarbose and Aging-Related Nutrient Signal Pathways (Figure 4)

The α-glucosidase inhibitors (AGI) are compounds that act primarily inside the gastrointestinal tract and inhibit the digestion of complex carbohydrates by α-aminoglycosides and α-amylase [76]. As an AGI, acarbose (HUADONG MEDICINE, Hangzhou, China) is mainly used to treat type 2 diabetes, hampering the breakdown and absorption of complex carbohydrates [77] and blocking postprandial glucose spikes [78]. However, a meta-analysis showed that a few patients have the gastrointestinal reactions in intaking acarbose [79]. Thus, acarbose as an anti-aging drug is uncertain, which needs more preclinical and clinical investigations to be confirmed.

Early in 2013, the Multicenter Intervention Testing Program (ITP) sponsored by the National Institute on Aging evaluated drugs that could increase healthy lifespan in genetically heterogeneous mice [80]. The studies showed that acarbose improved the healthspan and lifespan of aging *HET3 mice* [78]. The median lifespan of male mice treated with acarbose increased by approximately 20%, while the female mice increased by only 5% [81].

Researchers found that the halflife of acarbose was mainly related to short-chain fatty acids (SCFAs) in the acarbose and aging-related nutrient signal pathways (Figure 4). SCFAs, the main products of starch fermentation by intestinal bacteria, is beneficial for health. Acarbose changed the intestinal flora and its fermentation products, increasing the production of SCFAs by inhibiting host digestion, and facilitating the flux of starch to the lower digestive system, ultimately leading to prolonged lifespan [82].

In addition, ATF4 is also a key player in acarbose- and aging-related nutrient signal pathways (Figure 4). Current research revealed that the elevated ATF4 regulated cellular responses to changes in amino acid availability and ribosomal function through its DNA-binding activity, which were primarily responsible for the increased lifespan of mice [83].

Furthermore, age-related functional molecules also influence the process of aging, and are changed by acarbose in acarbose- and aging-related nutrient signal pathways (Figure 4). Yan et al. showed that long-term administration of acarbose alleviated age-related deficits in spatial learning and memory, associated with a higher level of the insulin/IGF-1 system and acetylated histone H4 lysine 8 (H4K8ac) versus the control *SAMP8 mice* [84]. When studying that long-term effects of treatment with acarbose on the behavioral defects and biochemical changes in *SAMP8 mice*, Tong et al. pointed out that these aging-related changes could be alleviated by acarbose treatment, particularly in terms of learning and memory capacity, which might be associated with homeostatic changes in the insulin/IGF-1 signal and glucose metabolism as well as in the levels of brain-derived neurotrophic factor (BDNF), IGF-1 receptors, and synaptotagmin 1 (Syt1) and syntaxin 1 (Stx1) [85].

The lifespan-prolonging mechanism of acarbose was proposed to be related to its potential mimetic effect of CR [76]. Dodds et al. showed that acarbose improved the survival of *Apc+/Min mice* by blocking the digestion of complex carbohydrates and by improving the metabolism of carbohydrate [86], which functions as a mimetic CR to regulate longevity.

Therefore, we believe that carbohydrates might undermine healthy longevity and increase mortality. Based on the efficacy of acarbose in diabetes, we think that an instant attention should be paid to the effects of hyperglycemia on aging diseases, such as cancer, and research on acarbose and other glucose-controlling drugs should be carried out.

**Figure 4 biomedicines-12-00127-f004:**
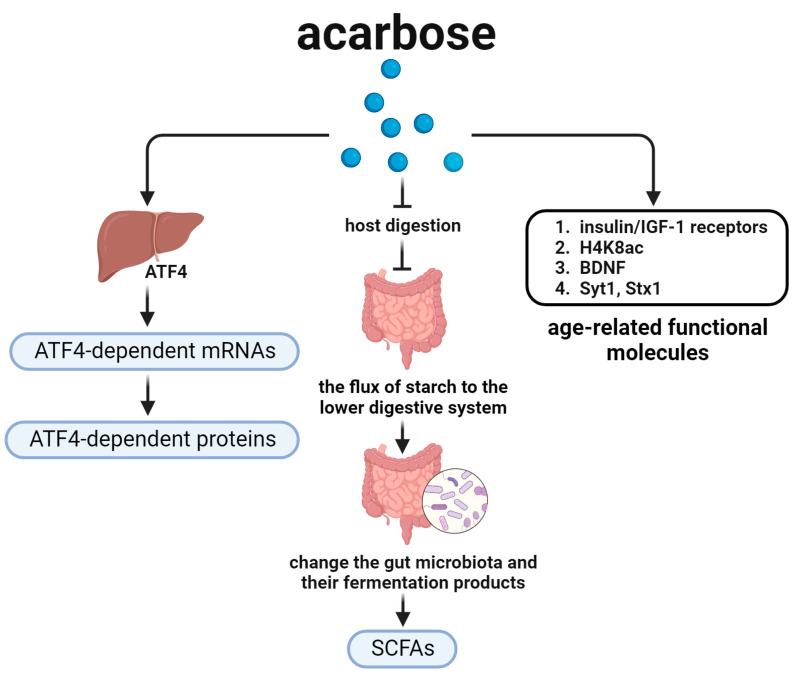
Acarbose and the aging-related nutrient signal pathways. This figure shows several signal pathways downstream of acarbose. Acarbose may change the gut microbiota, insulin/IGF-1 receptors, and ATF4 to regulate lifespan.

## 5. Nicotinamide Adenine Dinucleotide and the Aging-Related SIRT1 Signal Pathway (Figure 5)

NAD^+^ is commonly known as a coenzyme that mediates hundreds of redox reactions, which underlays a variety of processes [87,88]. The availability of NAD^+^ has been observed to diminish in geriatric populations and in instances of particular pathological states. Nicotinamide mononucleotide (NMN) (GeneHarbor, Yuyao, China) is a crucial intermediary in the production of NAD^+^. Numerous studies have demonstrated that NMN is capable of augmenting NAD^+^ biosynthesis and mitigating various pathological conditions [89,90]. NMN was shown to prevents age-related gene expression changes in key metabolic organs. For example, in skeletal muscle, NMN enhances oxidative metabolism in mitochondria and imbalances nucleus proteins, thereby effectively alleviating aging-related physiological decline in wild-type *C57BL/6N mice* [91].

The pro-lifespan effect of NAD^+^ was proposed to be attributed to a NAD^+^-dependent deacetylase, SIRT1 [92]. By conditional SIRT1 knockout mice, researchers found that SIRT1 presence is required for long lifespan. The elongated lifespan is concomitant with health advantages including attenuated liver steatosis, escalated insulin sensitivity, intensified locomotor activity, and standardization of gene expression patterns as well as indicators of inflammation and apoptosis [93].

Thus, it was speculated that NAD^+^ played a role in delaying aging primarily through the SIRT1 signal pathway, and this idea is confirmed by Verdin et al. [94]. In addition, researchers found that the overexpression of SIRT1 could reduce the accumulation of biotin associated with aging, delay the onset of age-related diseases, improve metabolism, and extend the lifespan in mammals [93,95]. SIRT1 may signal through several important downstream molecules, including NF-kB, FOXOs, PGC-1α, and PPARγ, as depicted in Figure 5 [96,97,98].

NF-kB is known as an inflammation molecule, which, however, plays a key role in aging process [96]. In aging-related diseases, SIRT1 activation would inhibit NF-kB, thereby decreasing the levels of inflammation to mitigate the disease process, thus helping extend lifespan [99,100]. Meanwhile, in cardiac myocytes, SIRT1-mediated activation of FOXOs could promote starvation-induced autophagic flux and left ventricular function maintenance to increase longevity [101]. Moreover, the pathway comprising PGC-1α and PPARγ is also regulated by SIRT1 [102]. In lipolysis, monounsaturated fatty acids (MUFAs) could activate SIRT1, which then signals to PGC-1α and PPARγ to increase mitochondrial biogenesis, finally resulting in delayed aging [102].

However, there may be a risk of tumorigenesis associated with NAD^+^ boosting [103]. As a high-dose supplementation greatly surpasses the body’s niacin requirements, there will be substantial clearance through urine. Nicotinamide clearance occurs through the process of methylation by nicotinamide N-methyltransferase, yet there is a lack of investigation into whether this leads to methyl donor depletion over time [103]. As a result, additional preclinical and clinical investigations are imperative to ascertain the enduring safety of NMN as potential therapeutic remedies for human subjects.

**Figure 5 biomedicines-12-00127-f005:**
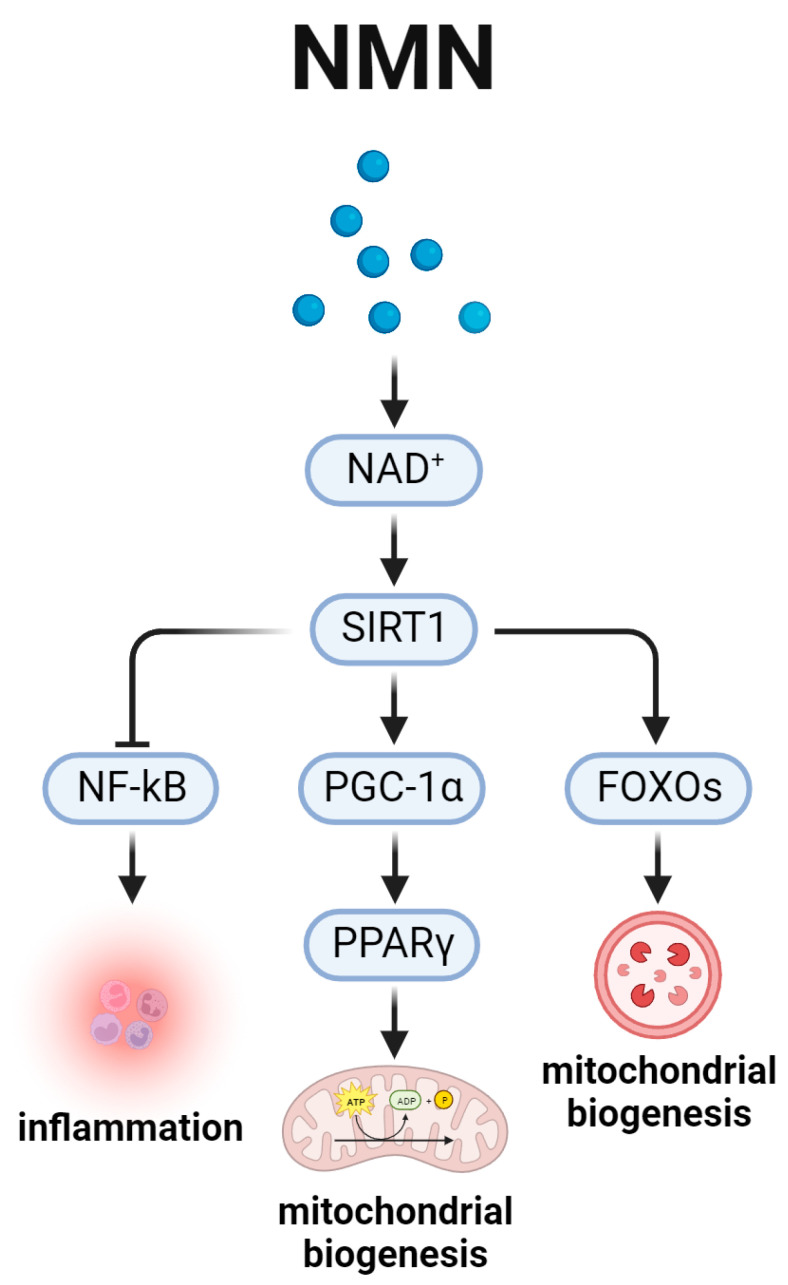
Nicotinamide adenine dinucleotide and the aging-related SIRT1 signal pathway. The figure shows several signal pathways, where NAD^+^ is a critical intermediate molecule. NAD^+^, and NMN as well, primarily functions through SIRT1 and its downstream molecules, including NF-kB to inhibit inflammation, PGC-1α/PPARγ to regulate mitochondrial biogenesis, and FOXOs to increase mitochondrial biogenesis, finally contributing to delayed aging.

## 6. Lithium and Telomere Length Regulation (Figure 6)

Lithium (Lithium Power, Perth, Australia) is a nutrient-essential trace element found mainly in vegetables, plant foods, and drinking water [104,105,106]. In clinical practice, lithium is mainly used for the prevention and treatment of bipolar disorder [107,108]. However, the toxicity of lithium is unignored. As early as 2012, McKnight et al. demonstrated that lithium is associated with not only renal function impairment, but also hypothyroidism, hyperparathyroidism, and weight gain. In consideration of its efficacy, McKnight et al. still suggest that lithium should be widely used [109].

In the treatment of patients with bipolar disorder, Pisanu et al. found that long-term lithium therapy prevented telomere shortening compared to the control group [110]. Telomere shortening is an important factor in the aging process. Therefore, Manrique et al. proposed that lithium might have a natural anti-aging effect [111]. This idea was confirmed by Zarse et al., who showed that low-dose lithium intake prolonged lifespan in humans [105]. This effect may be related with aging-related lithium pathway that signals though effectors including TERT, GSK-3β, FOXO1, and Akt (Figure 6) [112,113,114,115].

TERT is a key molecule to regulate longevity in lithium-related signal pathway. As a controller of telomere length, telomerase reverse transcriptase (TERT) could be modulated by lithium; that is, lithium could confer an anti-aging effect [116] by modulating TERT expression [112]. Consistently, lithium was found to be able to regulate histone methylation and chromatin structure to increase the survival of *C. elegans* [117].

GSK-3β is another essential molecule mediating the anti-aging effect of lithium. In a model of replicative senescence in human *WI-38 fibroblasts*, Zmijewski et al. found that lithium treatment reduced the aging-associated *p53* accumulation, and brought the cells into a reversible quiescent state. This effect was attributed to increased nuclear GSK-3β, which formed a complex with *p53* in nucleus. This association, in turn, promoted *p53* activation and contributed to cellular senescence [118].

As a regulator of *β-Catenin*, GSK-3β is involved in various pathways, neuro-degenerative diseases, and aging. Pons et al. demonstrated that compensatory increased gene expression of *β-Catenin* could rescue the effect of CSF1R deletion on *AD mice* [119]. It prompted us to realize that the relevant expression of GSK-3β/*β-Catenin* could change the process of age-related neurodegeneration. We presented such a hypothesis: lithium drove up p-GSK-3β, inhibited the activity of GSK-3β, encoded the expression of *β-Catenin*, and effected a determinate function in age-related neurodegeneration.

Akt and p-FOXO1 are also two important effectors in lithium-related signal pathway. In the cultures of aging cerebellar granule cells, lithium (10 mM) was able to exert anti-aging effects by activating Akt, which subsequently activated FOXO1 and inactivated GSK-3β by phosphorylation [120,121]. It is speculated that there is a link between GSK-3 inhibition and the regulation of genes responsible for telomere length, contributing to the anti-aging pharmacological effects.

**Figure 6 biomedicines-12-00127-f006:**
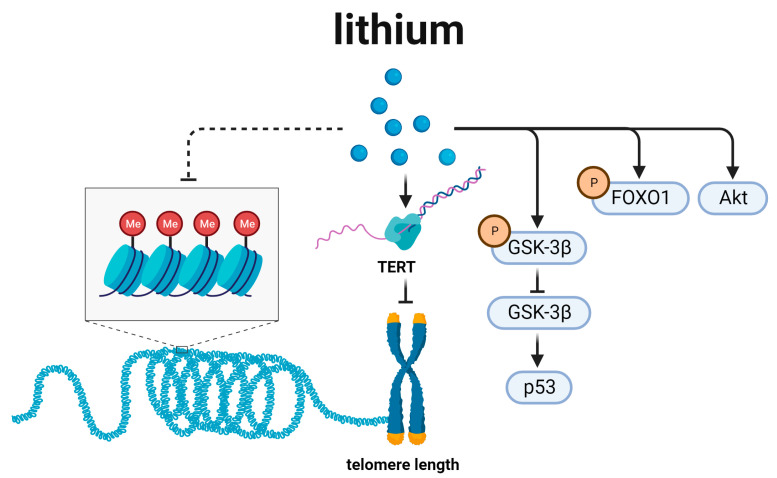
Lithium and the related signal pathways. TERT is a critical intermediate molecule in lithium signal pathway. Additionally, lithium also functions through GSK-3β, FOXO1, and Akt to regulate longevity.

## 7. NSAIDs and the Aging-Related Energy Metabolism (Figure 7)

Aspirin (Bayer AG, Leverkusen, Germany), a representative kind of NSAIDs, is a prototype cyclooxygenase inhibitor with various beneficial effects on human health. It prevents age-related diseases and delays aging [122,123]. An observational study in Finland indicated that daily intake of low-dose aspirin could prolong lifespan in the elderly [124]. The phenomenon might be related to the ability of aspirin to reduce irritation in the stomach and intestinal tract [125]. Aspirin is also capable of prolonging the lifespan of *C. elegans*, *Drosophila*, and mice [126,127,128]. The anti-aging effects of NSAIDs were believed to be mediated via PDK1, DAF-12/16, and inflammation (Figure 7).

The 3′-phosphoinositide-dependent kinase-1 (PDK1) is a component of the insulin/IGF-1 pathway, and could be targeted by NSAIDs to regulate longevity [129]. A study demonstrated that weakening the insulin-like pathway by aspirin was helpful to extend lifespan [130]. According to the oxidative damage theory, ROS constitute a major cause of aging, especially for molecular damage caused by super-oxides and their derivatives. This implies that reducing ROS levels could delay aging [131]. Consistently, recent studies showed that oxidative stress, modulated by insulin/IGF-1, was a major risk factor for age-related functional decline [132,133,134].

DAF-12/16 is also key to promote aging in NSAIDs-related signal pathway. In *C. elegans*, increasing metabolism in generative cells to activate downstream DAF-12/16 was an effective way to extend lifespan [135].

As described above, inflammation plays an important role in aging process, which is also functional in NSAIDs-related signal pathway. NSAIDs could reduce low-grade inflammation to restore muscle anabolism after meals in aged rats [136]. In addition, NSAIDs could also delay aging by preventing age-related brain atrophy [137].

Due to the wide range use of NSAIDs and their side effects, including increased adverse gastrointestinal, renal, and cardiovascular effects [138], there is a need for more clinical studies to explore the mechanisms and strategies that balance the anti-aging effects and side effects.

In summary, different anti-aging drugs may target different signal pathways to delay the aging process. Nevertheless, shared molecular targets were identified for different drugs, highlighting a central role of these molecules in the aging process. For example, FOXOs is a shared target of metformin, NMN, and lithium, PGC-1α transduces signals from both rapamycin and lithium, and Nrf2/SKN-1 is involved in both mTOR and AMPK signal pathways, which are targeted by rapamycin and metformin, respectively (Table 1). In this sense, attention should be paid to an additive effect surpassing the safety threshold when multiple drugs were taken. Meanwhile, it should be borne in mind that every anti-aging drug has its own adverse effect. For instance, rapamycin is immunosuppressive and metformin usually induces gastrointestinal reactions, which may prevent the public from sticking to a drug over the long term and should also be taken into account during combined drug use. It is noted that relatively less information on signal pathways has been reported for lithium, NMN, and NSAIDs in terms of aging, which prevents us from stimulating an insightful and comprehensive discussion on them.

**Figure 7 biomedicines-12-00127-f007:**
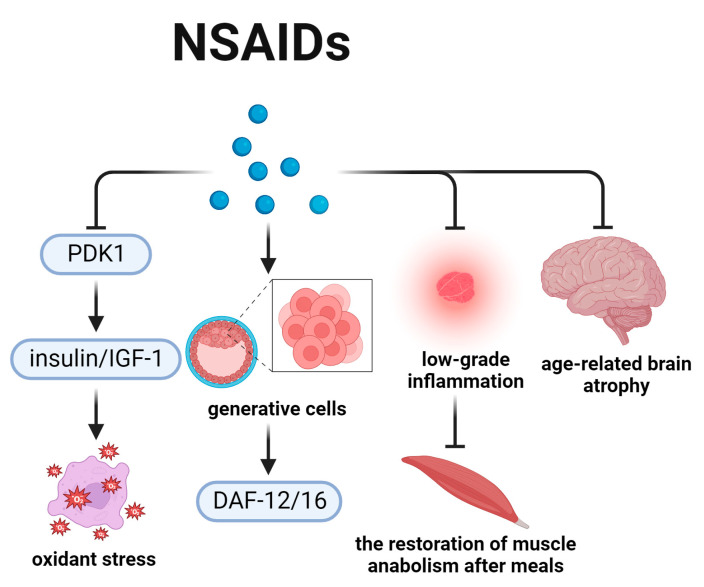
NSAIDs and the aging-related signal pathways. This figure shows that PDK1 is a critical intermediate molecule in NSAIDs aging-related signal pathway. Insulin/IGF-1 are primary downstream target molecules of PDK1. In addition, DAF-12/16 also plays a role in the process of aging. The anti-aging effects of NSAIDs are also involving the regulation of inflammation.

## 8. Conclusions

We conduct an overview on six major anti-aging drugs and their related signal pathways, which identified a significant cross talk across different signal pathways at both molecular and cellular levels, suggesting their pivotal roles in the aging process. Meanwhile, different drugs, to a greater or lesser extent, have their own drug-specific signal transduction, which provides an option for developing individualized anti-aging recipe through strategies like drug combination. For example, the combined use of acarbose along with NSAIDs or NMN might create a synergistic anti-aging effect, which remains to be explored in the near future.

## Figures and Tables

**Table 1 biomedicines-12-00127-t001:** The intersection of six anti-aging drugs and their signal pathways at molecular levels. The drugs are symbolized in the bottom right, the left column lists different molecular targets, the top row lists six signal pathways.

	mTOR	AMPK	Nutrient Signal Pathway	SIRT1	Telomere Length and GSK-3	Energy Metabolism		
Akt					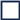			
ATF4			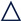					
BDNF			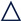					
DAF-12/16						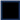		
DAF-16/FoxO	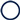							
ATP6AP1		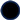						
NF-kB		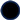		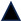				
Nrf2/SKN-1	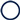	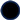						
H4K8ac			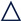					
HIF1α	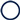							
IRE-1	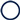							
SIRT1		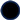		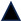				
CRTC/CREB								
p53		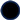			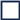			
p73	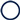							
PDK1						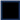		
PEN2		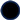						
TSC1/TSC2		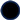						
mTOR	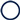	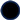						
ULK1	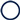	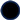						
S6K1	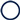	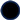						
4EBP1	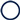	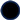						
FOXOs		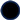		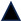	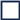		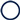	rapamycin
Syt1/Stx1			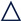				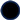	metformin
TERT					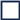		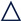	acarbose
PGC-1α	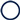			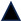			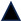	NMN
PPARγ				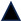			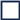	lithium
GSK-3β					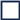		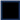	NSAIDS

## Data Availability

Data are contained within the article.

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
