# Peer review of "Anti-Aging Drugs and the Related Signal Pathways"

_biomedicines, 2024, doi:10.3390/biomedicines12010127_

Round 1
Reviewer 1 Report
Comments and Suggestions for Authors
The article titled: “Anti-aging drugs and related signaling pathways” by Nannan Du et al., covers an important topic of aging and drug development. However, there are several gaps in knowledge or definitions that need to be addressed, otherwise, the article is quite limited; and does not cover important topics on aging and drug development. Furthermore, all the compounds suggested by the authors as active against aging have already been published and are considered geroprotectors, so the present review does not add any new information in the field, nor does it even add information on geroprotectors. Another important issue is that there are no conclusions or discussions, it is just a write-up of various published results on different topics associated with aging and some drugs, add them so that they can add to the readers some clues about the development of anti-aging drugs.
Correct the font size format as there are several phrases with different sizes or formats. The same goes for handwriting and spelling, there are places in the manuscript without punctuation. Add figure legends to each figure; Otherwise, it will be quite difficult to follow the figure.Table doesn´t have any legend or description.
What do you mean by "point out important and critical pathways for the aging process"? What do you not consider epigenetic pathways, inflammation, senescence or dysbiosis? What is your definition of mechanism-based anti-aging strategies?
Introduction
Why is an aging graph proposed based on data estimated up to 2090? How do you consider or estimate such a prediction? Where do you get that information from? From the WHO?
When you state that "the phenomenon of DNA double-strand breakage
repair has been shown to be a new target for anti-aging interventions."
what do you mean? Please add information and references to your statements.
Lithium has been considered a fairly toxic compound, why isn't there some discussion about such a limitation?
Table 1 is quite confusing and very difficult to understand. Please improve this table.
Comments on the Quality of English LanguageCorrect the font size format as there are several phrases with different sizes or formats. The same goes for handwriting and spelling, there are places in the manuscript without punctuation. Add figure legends to each figure; Otherwise, it will be quite difficult to follow the figure.Table doesn´t have any legend or description.
Author Response
Re: Manuscript ID: 2667143 – Anti-aging drugs and the related signaling pathways submitted to the journal " biomedicines".
#Reviewer 1
Comments and Suggestions for Authors
The article titled: “Anti-aging drugs and related signaling pathways” by Nannan Du et al., covers an important topic of aging and drug development. However, there are several gaps in knowledge or definitions that need to be addressed, otherwise, the article is quite limited; and does not cover important topics on aging and drug development. Furthermore, all the compounds suggested by the authors as active against aging have already been published and are considered geroprotectors, so the present review does not add any new information in the field, nor does it even add information on geroprotectors. Another important issue is that there are no conclusions or discussions, it is just a write-up of various published results on different topics associated with aging and some drugs, add them so that they can add to the readers some clues about the development of anti-aging drugs.
Q1: Correct the font size format as there are several phrases with different sizes or formats. The same goes for handwriting and spelling, there are places in the manuscript without punctuation. Add figure legends to each figure; Otherwise, it will be quite difficult to follow the figure. Table doesn´t have any legend or description.
Response: Thanks for your reminder. We have corrected the font size format and added punctuation in the revised manuscript. Meanwhile, we have added figure legends to each figure (line 46 to 48, line 139 to 144, line 195 to 199, line 249 to 251, line 294 to 299, line 340 to 342, line 372 to 376), likewise, we have added legend to the table (line 378 to 380).
Q2: What do you mean by "point out important and critical pathways for the aging process"? What do you not consider epigenetic pathways, inflammation, senescence or dysbiosis? What is your definition of mechanism-based anti-aging strategies?
Response: Thanks for your question. About the "point out important and critical pathways for the aging process", we think in the aging process, anti-aging drugs will be very popular, and the use of anti-aging drugs will be quite extensive. Thus, the pathways of anti-aging drugs provide some cues, may be include important and critical pathways for the aging process. Aging is a very complicated and important process. There have been reported plenty of hallmarks, such as genomic instability, telomere attrition, epigenetic alterations, loss of proteostasis, disabled macroautophagy, deregulated nutrient-sensing, mitochondrial dysfunction, cellular senescence, stem cell exhaustion, altered intercellular communication, chronic inflammation and dysbiosis (DOI: 10.1016/j.cell.2022.11.001). Therefore, we also consider that epigenetic pathways, inflammation, senescence and dysbiosis are the driven factors in the process of aging. However, take the complexity of aging into account, we just focus on the six signaling pathways. As for the above factors, we will discuss them in the next work. As for the definition of mechanism-based anti-aging strategies, we think the six signaling pathways provide some cues that we can target some important and critical targets to exert anti-aging effects from the level of mechanisms.
Introduction
Q3: Why is an aging graph proposed based on data estimated up to 2090? How do you consider or estimate such a prediction? Where do you get that information from? From the WHO?
Response: Thank you for your reminder. The data source of aging graph is from the WHO (https://population.un.org/wpp/Download/Standard/MostUsed/), which demonstrate a prediction on age distribution of the population to show that there are more and more aging people. We consider such a prediction could be able to attract people's attention. It will make people consider the reason of the change on age distribution of the population, promote people to think the method to decelerate, stop, or reverse aging by therapeutic interventions. Thus, it will boost the field of process on aging.
Q4: When you state that "the phenomenon of DNA double-strand breakage repair has been shown to be a new target for anti-aging interventions." what do you mean? Please add information and references to your statements.
Response: Thank you for your question. In the panel of 18 rodent species with diverse lifespans, Tian et al. show that more robust DNA double-strand break repair coevolves with longevity (DOI: 10.1016/j.cell.2019.03.043). Therefore, we consider their conclusion provides the powerful evidence in anti-aging interventions. Thus, we state that "the phenomenon of DNA double-strand breakage repair has been shown to be a new target for anti-aging interventions".
Q5: Lithium has been considered a fairly toxic compound, why isn't there some discussion about such a limitation?
Response: Thank you for your suggestion. As you can see, the toxicity of lithium is impossible to ignore and attracts people a lot attention, though it has been regarded an anti-aging drug. As early as 2012, McKnight et al. demonstrate that lithium is not only associated with renal function, but hypothyroidism, hyperparathyroidism, and weight gain through a systematic review and meta-analysis (DOI: 10.1016/S0140-6736(11)61516-X). We have also added the limitation of lithium in the section of "Lithium and telomere length regulation". Please read it in line 303 to 306.
Q6: Table 1 is quite confusing and very difficult to understand. Please improve this table.
Response: Thank you for your advice. We have added the legend to the table. Take the example of rapamycin, the words of top line mean six signal pathways, the words of left line mean potential targets in table 1. Thus, the point means the intersection of targets and signal pathways, which could prompt us the potential critical targets and related signal pathways.
Thanks.
Reviewer 2 Report
Comments and Suggestions for Authors
"The manuscript "Anti-aging drugs and the related signaling pathways" is complete, well-written, and interesting. It brings new insight into the role of certain molecules in aging.
The figures are clear and not misleading.
I only have minor comments:
1- Please add a legend below each figure; it would help in understanding them.
2- You mentioned GSK-3Beta. The latter is a regulator of Beta-Catenin, which is involved in various pathways, neurodegenerative diseases, and aging (10.1186/s13195-020-00747-7). Beta-Catenin is also regulated by Oestrogens. Could you please discuss the role of GSK3-B and Beta-Catenin in aging and the potential effect of Lithium on this pathway?
3- It would be interesting to discuss the effect of these molecules on the immune system."
The changes made include adding commas for better readability, correcting some capitalization, and making minor adjustments for clarity.
Thanks
Author Response
Re: Manuscript ID: 2667143 – Anti-aging drugs and the related signaling pathways submitted to the journal " biomedicines".
#Reviewer 2
Comments and Suggestions for Authors
The manuscript "Anti-aging drugs and the related signaling pathways" is complete, well-written, and interesting. It brings new insight into the role of certain molecules in aging.
Response: Thank you for your comment.
The figures are clear and not misleading.
Response: Thank you.
I only have minor comments:
Q1: Please add a legend below each figure; it would help in understanding them.
Response: Thank you for your reminder.
The legend of figure 1 is: the age distribution predicted by WHO. In the figure 1, we dived population into 4 sections, 0-4 years old, 5-14 years old, 15-59 years old and 60+ years old. Since 2000, the 0-4 and 5-14 populations start to decrease while the 60+ population is increasing. We added them in manuscript, please read it in line 46 to 48.
The legend of figure 2 is: rapamycin and the aging-related mTOR signal pathway. This figure shows several signal pathways downstream of rapamycin. Rapamycin mainly functions through mTOR and their downstream molecules, such as DAF-16/FoxO and Nrf2/SKN-1, to regulate protein homeostasis, PGC-1α to influence mitochondrial respiration, ULK1 and p73 to regulate autophagy, and HIF1α to inhibit ER stress, aiming to regulate longevity. We added them in manuscript in line 139 to 144.
The legend of figure 3 is: metformin and the aging-related AMPK signal pathway. Metformin mainly functions through PEN2 to active ATP6AP1, thereby inducing AMPK-activation. Its downstream molecules include NF-kB to regulate inflammation, Nrf2/SKN-1, SIRT1, and TSC1/TSC2 to decrease cellular stress resistance, mTOR/S6K1 and mTOR/4EBP to regulate cell growth, ULK1 to activate autophagy. We added them in manuscript in line 195 to 199.
The legend of figure 4 is: acarbose and the aging-related nutrient signal pathways. This figure shows several signal pathways downstream of acarbose. Acarbose may change the gut microbiota, insulin/IGF-1 receptors and ATF4 to regulate lifespan. We added them in manuscript, please read it in line 249 to 251.
The legend of figure 5 is: nicotinamide adenine dinucleotide and the aging-related SIRT1 signal pathway. The figure shows several signal pathways, where NAD+ is a critical intermediate molecule. NAD+, and NMN as well, primarily functions through SIRT1 and its downstream molecules, including NF-kB to inhibit inflammation, PGC-1α/PPARγ to regulate mitochondrial biogenesis, and FOXOs to increase mitochondrial biogenesis, finally contributing to delayed aging. We added them in manuscript in line 294 to 299.
The legend of figure 6 is: lithium and the related signaling pathways. TERT is a critical intermediate molecule in lithium signal pathway. Additionally, lithium also functions through GSK-3β, FOXO1 and Akt to regulate longevity. We added them in manuscript, please read it in line 340 to 342.
The legend of figure 7 is: NSAIDs and the aging-related signal pathways. This figure shows that PDK1 is a critical intermediate molecule in NSAIDs’ aging-related signal pathway. Insulin/IGF-1 are primary downstream target molecules of PDK1. In addition, DAF-12/16 also plays a role in the process of aging. The anti-aging effects of NSAIDs are also involving the regulation of inflammation. We added them in manuscript in line 372 to 376.
The legend of figure 8 is: the intersection of six anti-aging drugs and their signal pathways at molecular levels. The drugs are symbolized in the right bottom, the left column lists different molecular targets, the top row lists six signal pathways. We added them in manuscript, please read it in line 378 to 380.
Q2: You mentioned GSK-3Beta. The latter is a regulator of Beta-Catenin, which is involved in various pathways, neurodegenerative diseases, and aging (10.1186/s13195-020-00747-7). Beta-Catenin is also regulated by Oestrogens. Could you please discuss the role of GSK3-B and Beta-Catenin in aging and the potential effect of Lithium on this pathway?
Response: Thank you for your suggestion. As inspired by the reference quoted by the reviewer, we discussed the role of GSK-3β and β-Catenin in aging, specifically the brain aging, and the potential effect of Lithium on this pathway. Please find the related part in the main text (Page9, line 326-332).
Q3: It would be interesting to discuss the effect of these molecules on the immune system.
Response: Thank you for your suggestion. As suggested, we discussed the effect of these molecules in the immune system with microglia, the innate immune cells, as the representative example in brain aging context, which was included into the main text (line 333-338).
Q4: The changes made include adding commas for better readability, correcting some capitalization, and making minor adjustments for clarity.
Response: Thank you for your advice. We have corrected them in the corresponding places in the manuscript.
Thanks.

Reviewer 3 Report
Comments and Suggestions for Authors
This review discusses about anti-aging drugs and the related signaling pathways. The authors discuss six types of drugs that target different pathways involved in aging, such as mTOR, AMPK, IGF-1, SIRT1, telomere length, and energy metabolism. The authors provide evidence from various animal models and human studies to support the anti-aging effects of these drugs.
Major points:
The article is well-referenced, giving a clear overview of the field.
The article covers a wide range of signaling pathways and drugs that are relevant to aging research and intervention.
The article provides a comprehensive overview of the current state of knowledge and challenges in the field of anti-aging pharmacology.
Minor points:
The article does not address the potential side effects, safety issues, or ethical implications of using anti-aging drugs in humans.
This point should be discussed in detail, as the topic is quite sensitive.
The article does not compare the relative efficacy or mechanisms of action of different anti-aging drugs or combinations of drugs. The summary at the end of the article, could be expanded, to give a general conclusion.
Comments on the Quality of English Languagefine minor spellcheck needed
Author Response
Re: Manuscript ID: 2667143 – Anti-aging drugs and the related signaling pathways submitted to the journal " biomedicines".
#Reviewer 3
Comments and Suggestions for Authors
This review discusses about anti-aging drugs and the related signaling pathways. The authors discuss six types of drugs that target different pathways involved in aging, such as mTOR, AMPK, IGF-1, SIRT1, telomere length, and energy metabolism. The authors provide evidence from various animal models and human studies to support the anti-aging effects of these drugs.
Response: Thank you.
Major points:
The article is well-referenced, giving a clear overview of the field.
Response: Thank you.
The article covers a wide range of signaling pathways and drugs that are relevant to aging research and intervention.
Response: Thanks.
The article provides a comprehensive overview of the current state of knowledge and challenges in the field of anti-aging pharmacology.
Response: Thanks.
Minor points:
Q1: The article does not address the potential side effects, safety issues, or ethical implications of using anti-aging drugs in humans. This point should be discussed in detail, as the topic is quite sensitive.
Response: Thank you for your reminder. We have added the discussion of potential side effects about rapamycin (line 130 to 134), metformin (line 187 to 193), acarbose (line 205 to 206), NMN (line 285 to 289), lithium (line 303 to 306), and NSAIDs (line 367 to 368), please see the detail in the revised manuscript.
Q2: The article does not compare the relative efficacy or mechanisms of action of different anti-aging drugs or combinations of drugs. The summary at the end of the article, could be expanded, to give a general conclusion.
Response: Thank you for your suggestion. Due to the limited number of relevant studies, fewer studies have been conducted on anti-aging drugs using the same model organism. On the PubMed website, we conducted a search for two relevant representative studies. One is a new synthetic drug, which is the combination of curcumin and aspirin (DOI: 10.3390/molecules26216609) to estimate the anti-aging efficiency in C. elegans, the other is a combination of two drugs (rapamycin and acarbose, DOI: 10.1111/acel.13724) to estimate the anti-aging efficiency in UM-HET3 mice. However, the two studies did not mention the combination of anti-aging drugs in natural aging mice, which is inconsistent with our theme. Therefore, our study did not compare the relative efficacy or mechanisms of action of different anti-aging drugs or combinations of drugs. But then, we will discuss this point further in our next study just like we have done with the anti-aging. We have expanded the summary at the end of the article, please read it in line 395 to 402.
Comments on the Quality of English Language
Q3: fine minor spellcheck needed.
Response: Thank you for your suggestion. We have corrected the minor spellcheck in the corresponding places of the revised manuscript.
Thanks.

Reviewer 4 Report
Comments and Suggestions for Authors
Abstract:
- The abstract is well-structured and provides a clear overview of the article. However, consider adding some information about the methods used in this review to provide a more complete summary.
Introduction:
- The introduction is informative and sets the stage for the article. However, consider breaking it into shorter paragraphs for better readability.
Section 2: Rapamycin and the mTOR signaling pathway:
- The section is detailed but complex. Consider breaking it into smaller subsections for better organization.
- The use of figures (Fig. 2) in this section is mentioned, but the figures themselves are not included in the text.
- A reference for the figures mentioned in the text should be included.
- While explaining scientific concepts, try to use simpler language where possible to improve readability for a broader audience.
- Clarify how each study cited relates to the main point of this section.
- For improved clarity, consider using a numbered list or bullet points for the various studies mentioned.
- Ensure consistent and clear use of abbreviations (e.g., DR, CR, ROS) and provide definitions when first used.
- In Figure 2's reference, make sure it corresponds to the figure discussed in this section.
Section 3: Metformin and the AMPK signaling pathway:
- Similar to the previous section, consider breaking this into smaller subsections.
- The references to specific figures (Fig. 3) should be correctly labeled and cited.
- Clarify how the studies presented relate to the main point of this section.
- Use simpler language and explain complex scientific terms for better accessibility.
Section 4: Acarbose and nutrient-related factor signaling pathways:
- This section follows the same pattern as the previous two sections. Break it into subsections for improved organization.
- Correctly label the references to figures (Fig. 4) and ensure they are cited accurately.
- Explain how the studies you mention are connected to the main subject of this section.
Section 5: Nicotinamide adenine dinucleotide and the SIRT1-related signaling pathway:
- Same as previous comments, break this section into subsections for clarity.
- Ensure accurate references to figures (Fig. 5) and clarify their relevance to the content.
- Explain the connection between the cited studies and the primary topic of this section.
Section 6: Lithium and telomere length regulation:
- Organize this section into subsections for clarity.
- Accurately reference the figures (Fig. 6) and explain their relevance.
- Use simpler language when discussing complex scientific ideas and ensure that the concepts are clearly explained
- Check for consistency in the use of abbreviations (e.g., SMCs).
Section 7: Aspirin and energy metabolism:
- Organize this section into subsections to improve readability.
- Accurately label references to figures and ensure they are properly cited.
- Explain the relevance of the studies to the main topic of this section.
General:
- Throughout the article, make sure to clarify how each study discussed is relevant to the primary subject matter of the section. This will help the reader understand the significance of the research.
- Consider improving the consistency of language and terminology throughout the article for better readability.
- Add a conclusion section to summarize the main findings and their implications.
- Proofread the article for grammar and punctuation errors.
Moderate editing of English language required
Author Response
Re: Manuscript ID: 2667143 – Anti-aging drugs and the related signaling pathways submitted to the journal " biomedicines".
#Reviewer 4
Comments and Suggestions for Authors
Abstract:
Q1: The abstract is well-structured and provides a clear overview of the article. However, consider adding some information about the methods used in this review to provide a more complete summary.
Response: Thank you for your suggestion. We have added some information about the methods used in this review in the manuscript. Please read it in line 13 to 15.
Introduction:
Q1: The introduction is informative and sets the stage for the article. However, consider breaking it into shorter paragraphs for better readability.
Response: Thank you for your suggestion. We have broken it into shorter paragraphs, please see the details in the revised manuscript.
Section 2: Rapamycin and the mTOR signaling pathway:
Q1: The section is detailed but complex. Consider breaking it into smaller subsections for better organization.
Response: Thank you for your advice. We have broken it into smaller subsections, and you can read it in the corresponding section.
Q2: The use of figures (Fig. 2) in this section is mentioned, but the figures themselves are not included in the text.
Response: Thank you for your reminder. Because Fig. 2 is edited by us, the figure included the corresponding section, we just cite it in the appropriate section.
Q3: A reference for the figures mentioned in the text should be included.
Response: Thanks for your suggestion. As we mentioned above, the figures were edited by ourselves, thus, we are unable to cite a reference for the figures.
Q4: While explaining scientific concepts, try to use simpler language where possible to improve readability for a broader audience.
Response: Thank you for your reminder. We have corrected the mentioned in the revised manuscript, and thank you for your reminder again.
Q5: Clarify how each study cited relates to the main point of this section.
Response: Thanks for your suggestion. We have clarified the relation in the corresponding section, please see the details in the revised manuscript.
Q6: For improved clarity, consider using a numbered list or bullet points for the various studies mentioned.
Response: We thank the reviewer for the constructive advice. Yes, as you may find in the revised manuscript, substantial efforts were made to increase the clarity, including fragmenting the paragraphs, rephrasing the sentences, cutting out unnecessary description, restructuring part of narration and redrawing the illuminations, etc. We wish that the reviewer may find the revised manuscript significantly improved in clarity. Thanks.
- Ensure consistent and clear use of abbreviations (e.g., DR, CR, ROS) and provide definitions when first used.
Response: Thank you for your reminder. We have ensured the consistent and clear use of abbreviations, and provided definitions when they were first used.
- In Figure 2's reference, make sure it corresponds to the figure discussed in this section.
Response: Thank you for your advice. We have checked the point, and we make sure it corresponds to the figure discussed in this section.
Section 3: Metformin and the AMPK signaling pathway:
Q1: Similar to the previous section, consider breaking this into smaller subsections.
Response: Thank you for your advice. We have broken it into smaller subsections in the revised manuscript, please see the details in the corresponding section.
Q2: The references to specific figures (Fig. 3) should be correctly labeled and cited.
Response: Thank you for your reminder. Because Fig. 3 is edited by us, the figure included the most corresponding section, and we just cite it in the appropriate section.
Q3: Clarify how the studies presented relate to the main point of this section.
Response: Thanks for your suggestion. We have clarified the relation in the corresponding section, please see the details in the revised manuscript.
Q4: Use simpler language and explain complex scientific terms for better accessibility.
Response: Thank you for your reminder. We have used simpler language and explained complex scientific terms in line with the corresponding cited reference. And we have corrected the mentioned in the revised manuscript.
Section 4: Acarbose and nutrient-related factor signaling pathways:
Q1: This section follows the same pattern as the previous two sections. Break it into subsections for improved organization.
Response: Thank you for your advice. We have broken it into subsections in the corresponding section, please see the details in the revised manuscript.
Q2: Correctly label the references to figures (Fig. 4) and ensure they are cited accurately.
Response: Thank you for your reminder. Because Fig. 4 is edited by us, the figure included the most corresponding section, we just cite it in the appropriate section.
Q3: Explain how the studies you mention are connected to the main subject of this section.
Response: Thanks for your suggestion. We have explained the connection in the corresponding section, please see the details in the revised manuscript.
Section 5: Nicotinamide adenine dinucleotide and the SIRT1-related signaling pathway:
Q1: Same as previous comments, break this section into subsections for clarity.
Response: Thank you for your advice. We have broken it into smaller subsections in the revised manuscript, please read it in the corresponding section.
Q2: Ensure accurate references to figures (Fig. 5) and clarify their relevance to the content.
Response: Thank you for your reminder. Because Fig. 5 is edited by us, the figure included the most corresponding section, we just cite it in the appropriate section.
Q3: Explain the connection between the cited studies and the primary topic of this section.
Response: Thanks for your suggestion. We have explained the connection in the revised manuscript, please see the details in the corresponding section.
Section 6: Lithium and telomere length regulation:
Q1: Organize this section into subsections for clarity.
Response: Thank you for your advice. We have broken it into subsections in the corresponding section, please read it in the revised manuscript.
Q2: Accurately reference the figures (Fig. 6) and explain their relevance.
Response: Thank you for your reminder. Because Fig. 6 is edited by us, the figure included the most corresponding section, we just cite it in the appropriate section.
Q3: Use simpler language when discussing complex scientific ideas and ensure that the concepts are clearly explained.
Response: Thank you for your reminder. We have used simpler language and ensured that the concepts were clearly explained in line with the corresponding cited reference. Besides, we have corrected the mentioned in the revised manuscript.
Q4: Check for consistency in the use of abbreviations (e.g., SMCs).
Response: Thank you for your suggestion. We have checked the consistency in the use of abbreviations. Pleas read it in the revised manuscript.
Section 7: Aspirin and energy metabolism:
Q1: Organize this section into subsections to improve readability.
Response: Thank you for your advice. We have broken it into smaller subsections in the corresponding section, please see the details in the revised manuscript.
Q2: Accurately label references to figures and ensure they are properly cited.
Response: Thank you for your reminder. We have accurately labelled references to figures and ensured they were properly cited. Please read the details in the revised manuscript.
Q3: Explain the relevance of the studies to the main topic of this section.
Response: Thanks for your suggestion. We have explained the relevance in the revised manuscript, please see the details in the revised manuscript.
General:
- Throughout the article, make sure to clarify how each study discussed is relevant to the primary subject matter of the section. This will help the reader understand the significance of the research.
Response: Thanks for your suggestion. We have clarified the relevance in the revised manuscript about the corresponding section, please see the details in the revised manuscript.
- Consider improving the consistency of language and terminology throughout the article for better readability.
Response: Thank you for your reminder. We have improved the consistency of language and terminology throughout the article in line with the corresponding cited reference for better readability. Besides, we have corrected the mentioned in the revised manuscript, please read the details in the corresponding section.
Q3: Add a conclusion section to summarize the main findings and their implications.
Response: Thanks for your advice. We have added a conclusion section to summarize the main findings and their implications, please see it line 395 to 402 of the revised manuscript.
Q4: Proofread the article for grammar and punctuation errors.
Response: Thanks for your reminder. We have proofread the grammar and punctuation errors in the revised manuscript.
Comments on the Quality of English Language
Moderate editing of English language required
Response: Thank you for your suggestion. We have made the corresponding editing of English language.
Thanks.
